# A Case Study on Structural Serviceability Subjected to Railway-Induced Vibrations at TOD Developed Metro Depot

**Yingying Liao [1,2], Peijie Zhang [2], Qiong Wu [3] and Hougui Zhang [1,3,*]**

1   State Key Laboratory of Mechanical Behavior and System Safety of Traffic Engineering Structures, Shijiazhuang Tiedao University, Shijiazhuang 050043, China; sjzlyy820@163.com
2   School of Civil Engineering, Shijiazhuang Tiedao University, Shijiazhuang 050043, China; 1202001104@student.stdu.edu.cn
3   Institute of Urban Safety and Environment Science, Beijing Academy of Science and Technology, Beijing 100054, China; 18600053885@163.com
*   Correspondence: zhanghougui@bmilp.com

**Abstract:** As a sustainable mode of metro-development strategy, transit-oriented development (TOD) is rapidly growing to finance the transport infrastructure investment. The main negative consequence of constructing residential buildings directly over metro depots is railway-induced vibration, that may affect structural serviceability. The residents may feel uncomfortable, as the metro trains start running very early in the morning and finish daily operations very late at night. In order to evaluate the level of human comfort subject to the special situation, a case study was provided in this paper. Directed by the academic review, there were four common comfort evaluation methods, with difference indexes to describe the influence of vibrations. Therefore, a measurement campaign was conducted and both acceleration and velocity sensors were simultaneously installed at the same measurement points, to reduce the influence of the conversion accuracy. The results show that there are certain differences between the evaluation methods in assessing the vibration comfort, but considering the most adverse effects together, the over-track building at this particular TOD-developed depot can ensure that 90% of the occupants would not be highly annoyed by the vibrations. The main negative effect on human comfort at the TOD depot is that the high-level vibrations would cause interruptions in sleep. Among them, the vibrations in this case would affect the rest of 17% of the occupants in the bedrooms on the seventh floor, and make it difficult for 9% of the occupants to fall asleep. Therefore, the evaluation index was suggested to consider more factors related to sleep difficulties and awake threshold values.

**Keywords:** structural serviceability; human comfort; metro railway vibrations; TOD development; sleep disturbance

## 1. Introduction

The development of urban rail transit systems affects, to a certain extent, the development level of cities. According to the information released by the China Urban Rail Transit Association [1], by the end of 2021, an urban rail transit system had been built and put into operation in a total of 50 cities in China, with a total length of 9192.62 km, including 7253.73 km of subways, accounting for 78.9% of the total rail transit system length. A metro depot is a basic ancillary facility of a metro system, that is used for the storage, cleaning, maintenance, and performance test of subway trains, and which usually covers a large land area [2]. With the large-scale construction and development of a subway system, a metro depot with a low building density and large floor space has not been an economical use of urban land [3]. In recent years, under the guidance of the transit-oriented development (TOD) model [4,5], many cities have started to develop over-track buildings at metro depots, which not only improved the urban land utilization, but also compensated for the deficits of the construction and operation of a metro.

The emergence of the over-track buildings has allowed the traditional roof of the metro depot to be replaced by a large reinforced concrete platform. The platform is supported by columns in the middle of the train tracks and divided horizontally, according to construction joints, to accommodate thermal expansion and create a modular aseismic structure [3]. The buildings located on these platforms are designed as multistoried or high-rise buildings and are intended for residential and commercial activities, such as residences, schools, stores, and restaurants. Due to the comprehensive advantages of high land-use efficiency, the short walking distance to subway stations, and high return on investment, these buildings are increasingly favored by designers and investors [6]. The vibration characteristics of a metro depot are related to many factors, such as speed, track type, sleeper, ballast, roadbed, building foundation, and structure [7,8]. However, the vibrations of the train power system and track structure and the dynamic interaction of the wheel-rail and the wheel-rail's unevenness are the main sources of vibrations in a track structure [9]. As a result of slow running, the vibrations generated by trains have been reduced, providing the possibility of building the over-track buildings above the metro depots. Unlike ordinary railroads, subways operate intensively at night and in the early morning. However, during the operation of a metro depot, the vibrations are transmitted without soil attenuation, thus having a large influence on the over-track buildings. The vibrations caused by extreme events, such as earthquakes, can generate large amplitudes and damage the safety of buildings; in contrast, train-induced building vibrations bring smaller amplitudes and do not affect building safety, but can cause discomfort to the occupants [10]. In recent studies, field measurements of the effects of train-induced vibrations caused by train operation on over-track buildings have been conducted. The results have shown that train-induced vibrations can be transmitted directly through the columns to the platform, and subsequently to the over-track buildings, thus possibly causing annoyance to the occupants [3,11–13].

Railway vibrations will have an impact on the comfort of the human body, and the subject has gradually attracted people's attention in recent years. As a result, the assessment of railway vibrations has become more common. [14]. The Chinese-issued vibration standards have often been used by developers as a control target. However, many of the projects that comply with vibration standards are still receiving a large number of complaints from occupants. Moreover, many foreign studies have shown that the occupants of buildings disturbed by vibrations cannot live freely, although the vibrations do not exceed the vibration standards [15]. The reason for such a situation is that the quality of living is not related to vibration indicators, but it is directly related to the comfort of the occupants. In China, research on the effect of vibrations on human comfort has been insufficient; also, there have been fewer application cases of the results of the vibration comfort research to the over-track building design in foreign countries. Aiming to improve the living quality and reduce the complaints of occupants, this paper focuses on the main factors of discomfort caused by the structural vibrations of the over-track buildings and provides a case reference for studying the effect of vibrations on human comfort in the over-track buildings.

The vibrations and noise generated by a railroad during its movement can affect the comfort of the occupants in the surrounding buildings, and this effect must be considered in the development of new lines or the reconstruction of the existing lines. Compared to noise, vibrations are often overlooked. However, due to an increase in public awareness and the success of noise mitigation measures, vibrations have become an increasingly important issue [16]. Human responses to railroad-induced vibrations include sleep disturbance, annoyance, and non-vibration factors.

In recent years, many studies on the effects of vibrations on human comfort were conducted, and significant results were achieved. The methods used to study the effects of vibrations on sleep include both objective and subjective measurements of sleep disturbance. The objective measurements of sleep disorders were mainly conducted by polysomnography (PSG), while subjective measurements of sleep disorders were usually performed

through questionnaires. Research has shown that vibrations can adversely affect the sleep quality of occupants. Arnsberg et al. [17] simulated the vibrations of heavy traffic and found that the vibrations can cause changes in sleep architecture and a reduction in rapid eye movement sleep. It has also been found that increased vibrations can increase the probability of waking up during the night and early morning [18], that is, it was demonstrated that the occupants can distinguish between train-induced vibrations and noise, and as the vibration amplitude increased, occupants' heart rate amplitudes and sleep disturbances increased, and sleep quality decreased [19]. It has been known that the vibrations caused by freight trains can increase the heart rate of people who are sleeping and may affect the cardiovascular function of occupants near the railroad [20]. The number of trains passing through a metro depot and the amplitude of induced vibrations have a negative effect on the sleep macrostructure, that is, a large number of trains and high vibration conditions increase the occurrence of sleep depth changes in the occupants, interrupt the continuity of slow-wave sleep, and increase the number of night-time awakenings [21]. The effects of traffic-induced vibrations on sleep are summarized in Table 1.

**Table 1.** Summary of the effects of traffic-induced vibrations on sleep [16].

| | Effect | Significant Findings |
|---|---|---|
| Biological changes | Change in cardiovascular activity | Increase in heart rate [19,20] |
| | Change in sleep structure | Reduction in REM sleep [17] |
| | | Greater number of sleep stage shifts [22] |
| | | Shorter period between falling asleep and first awakening [22] |
| | | Shorter maximum length of uninterrupted time spent in slow wave sleep [22] |
| | EEG awakening | Increase in probability of EEG awakening [22] |
| Sleep quality | Waking in the night/too early | Increase of reported awakenings/waking too early (Figure E.2 in [18]) |
| | Difficulty in getting back to sleep | Greater difficulty in getting back to sleep once awoken for higher amplitudes of vibration [22] |
| | Self-reported sleep disturbance from vibrations | Increase in proportion of people reporting sleep disturbances (Figure E.2 in [18] and Figure E.3 in [23]) |
| | | Self-reported sleep disturbances related to vibration amplitude [22] |
| | | Decrease in self-reported sleep quality [22] |
| | Self-reported sleep disturbances from noise | Vibration related to increase in proportion of people reporting sleep disturbances from noise [24] |
| | Decreased restoration | Decrease in self-reported restoration [22] |

Guski et al. [25] identified that annoyance is associated with disturbance, aggravation, dissatisfaction, concern, bother, displeasure, harassment, irritation, nuisance, vexation, exasperation, discomfort, uneasiness, distress, and hate. According to the EU FP7 project, CargoVibes, annoyance is a concept that has been widely used to evaluate the negative effect of environmental stressors on a population. It is a broad concept that describes the negative effects of vibrations on the environment from three aspects: activity disturbance, emotional responses, and attitudinal responses to the source of the annoyance. The effect of vibrations on people's annoyance is usually examined through questionnaires and field tests. It should be noted that the subjective responses of people to vibrations are significantly influenced by individual differences; therefore, the reference significance of individual responses to vibrations is not high, and only the proportion of people's responses to vibrations obtained by statistical laws on a large number of samples is valuable for studying the relationship between the annoyance rate and the vibration intensity. The EU FP7 project, CargoVibes, collected data from social vibration surveys conducted in seven countries. The collected data included 4490 samples. By analyzing these data, the curves of people's annoyance caused by railroad vibrations were plotted. These experimental data will be used in this study to evaluate the annoyance rate due to train-induced vibrations.

In addition, non-exposure factors cause vibrations to have an impact on human comfort. Table 2 summarizes the existing studies on the effect of non-exposure factors on people's annoyance. For the over-track buildings, the vibrations are generated at the arrival and departure times of metro trains, which are mostly arranged in the evening and early morning hours. Since occupants can commonly see a metro depot through the windows, the vibrations' source can be considered visible. The presence of the non-exposure factors will further increase the impact of vibrations on the occupants' comfort.

**Table 2.** Summary of the effects of the situation, attitudinal, and socio-demographic factors on annoyance [16].

| | Factor | Significant Findings |
|---|---|---|
| Time of day | Evening | Annoyance greater during the evening than during the day at the same level of vibration exposure [26] |
| | Night | Annoyance is greater during the night than during the day and evening at the same level of vibration exposure [26] |
| Situational | Situational | Annoyance greater if the source is visible [27,28] |
| | Time spent at home | Annoyance greater for people who spend less than 10 h per day at home [27] |
| | Type of area | Annoyance greater for people living in rural areas [27] |
| Attitudinal | Concern of damage | Annoyance greater for those concerned that vibrations are damaging their property or belongings [26,28] |
| | Expectation regarding future vibrations | Annoyance greater for those expecting vibrations to get worse in the future [27] |
| | Necessity of source | Annoyance greater for those considering the source unnecessary [28] |
| | Noise sensitivity | Annoyance from vibrations greater for those considering themselves as noise sensitive [28] |
| Sociodemographic | Age | Annoyance greater for those in the middle age group in [26], no significant effect in [28] |

In studies on vibration comfort, it is usually necessary to collect a large amount of actual measurement data and conduct a large number of questionnaire surveys, which is time-consuming and costly. Recently, some countries and institutions have fitted vibration–response curves and used them in studies on railroad line-induced vibrations. However, there have been fewer application cases of the research results for studying the vibration effect on over-track buildings.

This study provides the field test data of the over-track buildings in China and uses the exposure–response curve to analyze the vibrations impact on occupants' comfort in over-track buildings. The results presented in this study can help to develop strategies for providing better occupant comfort under train-induced vibrations and possible vibration reduction measures, which can help to improve the quality of living.

## 2. Description of Metro Depot and Over-Track Building and Vibration Measurement

The over-track buildings considered in this study are located above the operation depot of a metro depot. The over-track buildings consist of ten 11-storey residential buildings with a kindergarten and a shopping center. The number of available households in the over-track buildings is 613, and the total construction area is 111,311.31 m$^2$ with a frame structure system.

The plan view of the metro depot is shown in Figure 1. The north side of the metro depot is the maintenance depot, responsible for the daily maintenance of trains. On the south side of the metro depot is the testing line for the high-speed testing and performance evaluation of trains to ensure safe operation. The throat area is located on the west side of the operation depot, connecting the train entry and exit lines. The over-track building under the test is located above the operating depot, and the test tracks are 14–18. Table 3 shows

the train and track parameters in the operating metro depot. We measured the combined stiffness of the type I separate fastener system (see Figure 2). The average measurement results show that the combined static stiffness of the fastener is 40.35 KN/mm, and the calculated combined dynamic stiffness is 38.9~43.7 KN/mm.

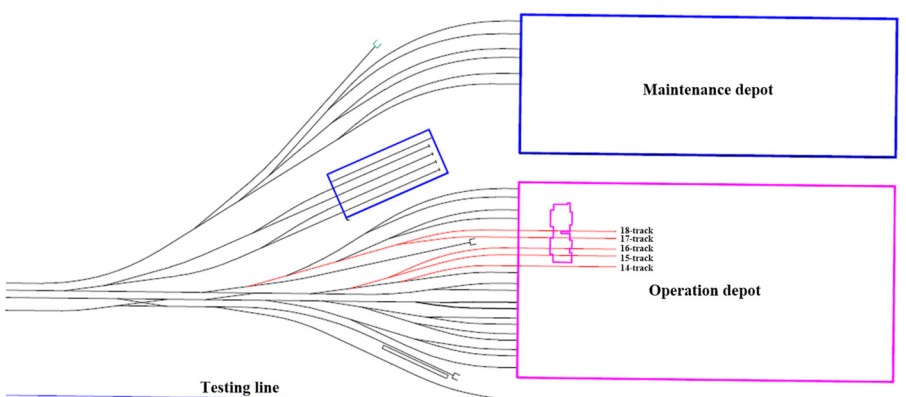

**Figure 1.** The plan view of the presented depot.

**Table 3.** Train and track parameters in the operating metro depot.

| Rail Types | Rail Weight | Train Speed | Fastener |
|---|---|---|---|
| Long sleeper embedded ballast less track | 60 kg/m | The warehouse door-10 km/h The middle of the operating depot-5 km/h | Type I separate fastener |

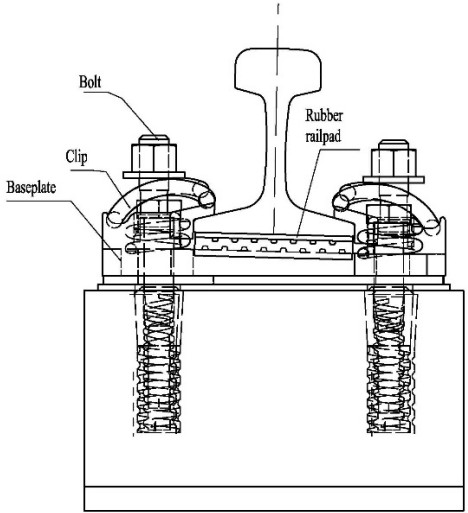

**Figure 2.** Type I separate fastener.

Since the speed of the train is the highest at the entrance/exit, the train will generate more vibrations in the building near the entrance/exit when the train is running, which will bring higher annoyance to the occupants. Therefore, the 11-story building in Figure 1 was selected as the test. Figure 3 shows the sectional view of the over-track building under the test, which has 11 floors; the test floors included the third, fifth, seventh, ninth, and 11th floors. The test floors are all affected by the vibrations of trains on tracks 14–18 under the platform.

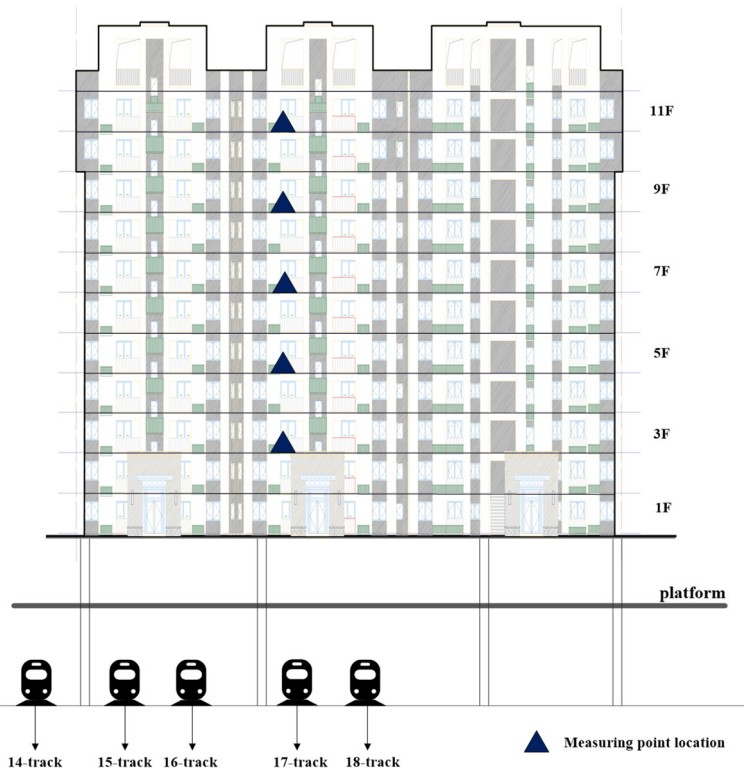

**Figure 3.** The sectional view of the over-track building.

To consider the vibration comfort in the rooms with different usage functions, field experiments were conducted to measure the amplitude in the drawing room and bedroom of each test floor, as shown in Figure 4. When a train passes, the vertical vibrations in a building are significantly greater than the horizontal vibrations [11]. Considering that the impact of vibration intensity and human comfort are directly negatively correlated, the vibration measurement and analysis considered only the vertical vibrations in this study. The test conditions were divided into two groups of working conditions: normal operation and scheduled shunting. The normal operation conditions included four peak-hour periods: 10:30–11:30 p.m.; 11:30–12:30 p.m.; 4:30–5:30 a.m.; and 5:30–6:30 a.m. Scheduled shunting was performed during the daytime to allow trains to pass through the test tracks as scheduled. Figure 5 shows the operational depot used in the field test.

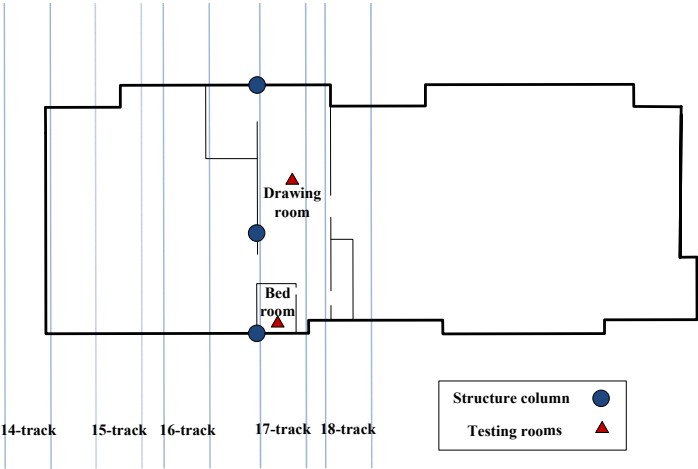

**Figure 4.** Measuring point location plan.

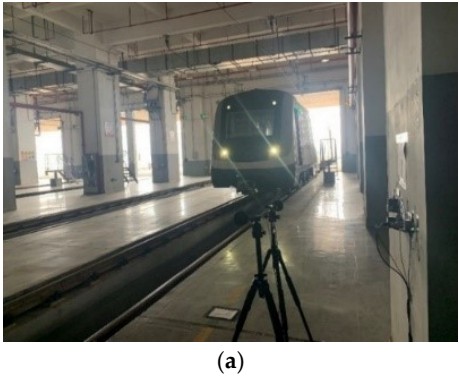 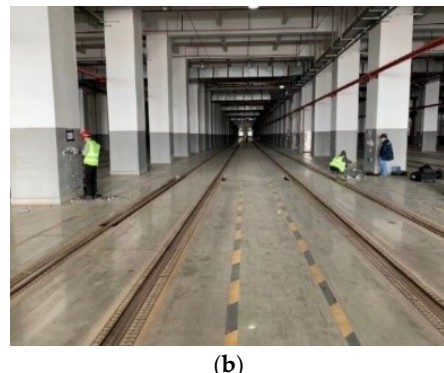

(**a**)                                                        (**b**)

**Figure 5.** Field testing in the operating metro depot. (**a**) Test train; (**b**) Test tracks.

The test train in this research is a B-type metro train with six cars and has a length of 118 m. In the case of no-load, the mass of the train is 202 t, and the axle load is less than 14 t. Figure 6 shows the instruments used in the measurement, which included the INV3062C1 data acquisition and the signal processing systems (China Orient Institute of Noise and Vibration, Beijing, China) used to collect the data on eight channels simultaneously. Since the acceleration and velocity were used as evaluation indexes of the vibration effect on human comfort, accelerometers and velocimeters were used for the measurements. The B&K8344 accelerometer with a sensitivity of 5 mV/g and the pickup 941B were installed at the target location. The pickup 941B contained the velocity and acceleration gears, and the velocity gear was used in the test. The B&K8344 accelerometer, the pickup 941B, and the acquisition system were calibrated before the test. A sampling rate of 2048 Hz at Nyquist frequency provided a meaningful level of spectrum below 1024 Hz. This sampling rate provided a large enough range to include the dominant frequencies for analysis.

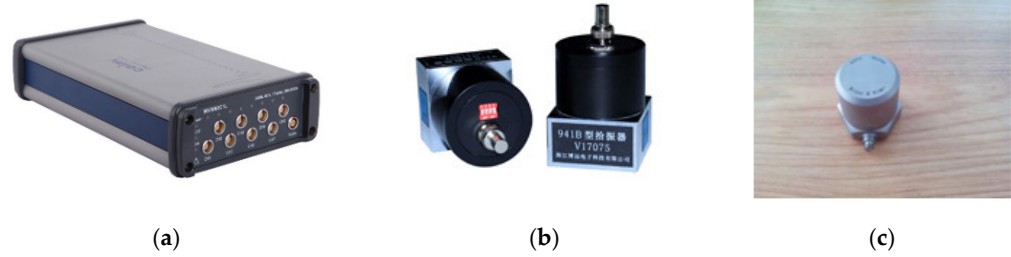

(**a**)                                     (**b**)                                     (**c**)

**Figure 6.** (**a**) INV3062C1 data acquisition; (**b**) Pickup 914B; (**c**) B&K8344 accelerometer.

## 3. Vibration Response Result Analysis

The collected data were analyzed, using the DASP analysis system and MATLAB software, to obtain the vibration response in the considered area.

### 3.1. Track Effect on Vibration Response

We selected the third-floor drawing room for analysis. The vibration response of the measured area of the building was evaluated when the train passed over the different tracks. As mentioned above, the test tracks included five tracks, 14–18.

The vibration responses of the accelerometer and velocity sensor in the time domain for the selected drawing room for a train passing over track 16 are shown in Figure 7. As shown in Figure 7, in the time–domain diagram, there was a clear spindle-shaped waveform at both ends, because the unified track was divided into two sections with a clear speed limit between them, that is, the speed was at first fast and then slow on the entry of track 16, but it was at first slow and then fast on the exit of track 16. The velocity and acceleration of the test building exhibited the same trend of vibration variation in the

time domain. When the train speed is higher, the vibration generated in the building is also higher.

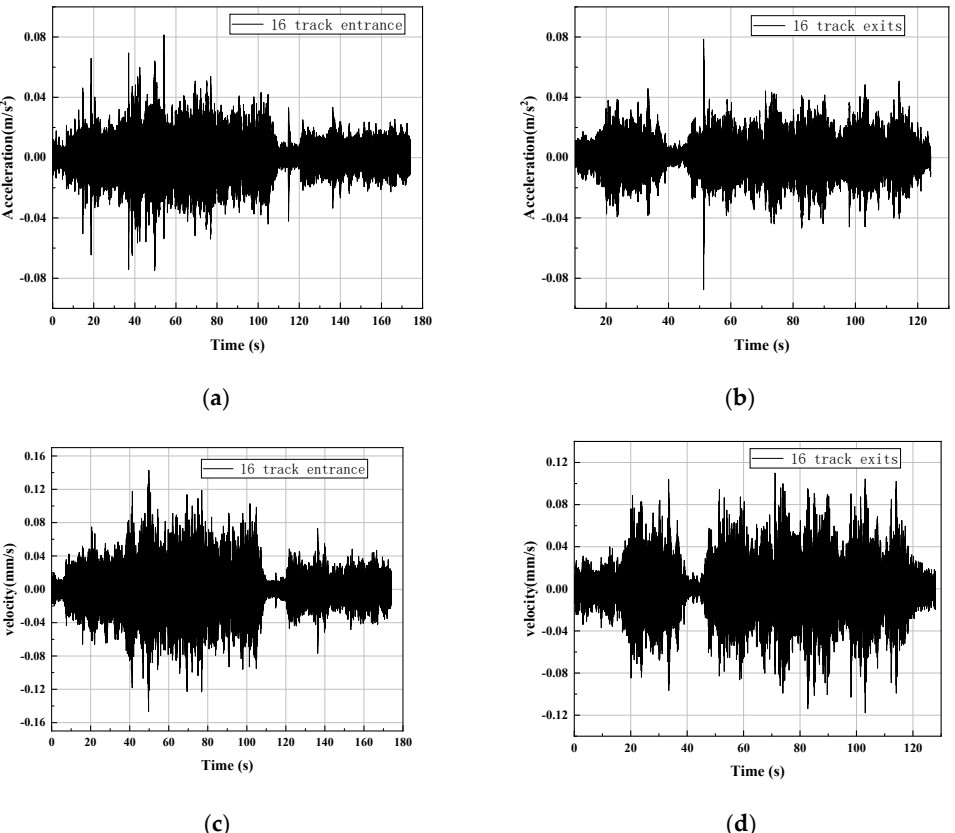

**Figure 7.** (**a**) Vibration acceleration response of the third-floor drawing room at the entry of track 16; (**b**) vibration acceleration response of the third-floor drawing room at the exits of track 16; (**c**) vibration velocity response of the third-floor drawing room at the entry of track 16; (**d**) vibration velocity response of the third-floor drawing room at the exits of track 16.

Figure 8 shows how the frequency spectrum and one-third octave spectra of the vertical vibrations of the drawing-room acceleration and velocity varied among the tracks; the reference velocity was $2.54 \times 10^{-8}$ m/s, and the reference acceleration was $1 \times 10^{-6}$ m/s$^2$.

The results show that for the different tracks, the vibrations transmitted to the building showed a peak of 30 Hz at both acceleration and velocity, with the main frequencies mostly between 20 Hz and 60 Hz. In the one-third octave spectrum, the velocity and acceleration followed the same trend in frequency, having a peak at 31.5 Hz. The peak frequency was related to the resonance of the vibration amplification in the corresponding frequency band with the vertical vibrations in a specific room. The ambient vibrations had a lower acceleration and velocity than the over-travel vibrations in the frequency domain, but the peak frequency was the same as that of the over-travel vibrations. When the train passed over different tracks, the vibrations in the building showed the trend that the vibrations of the vertically downward track of the tested room was larger than the non-vertically downward track vibrations, that is, the vibrations on tracks 14 and 15 were smaller than the vibrations on tracks 16–18.

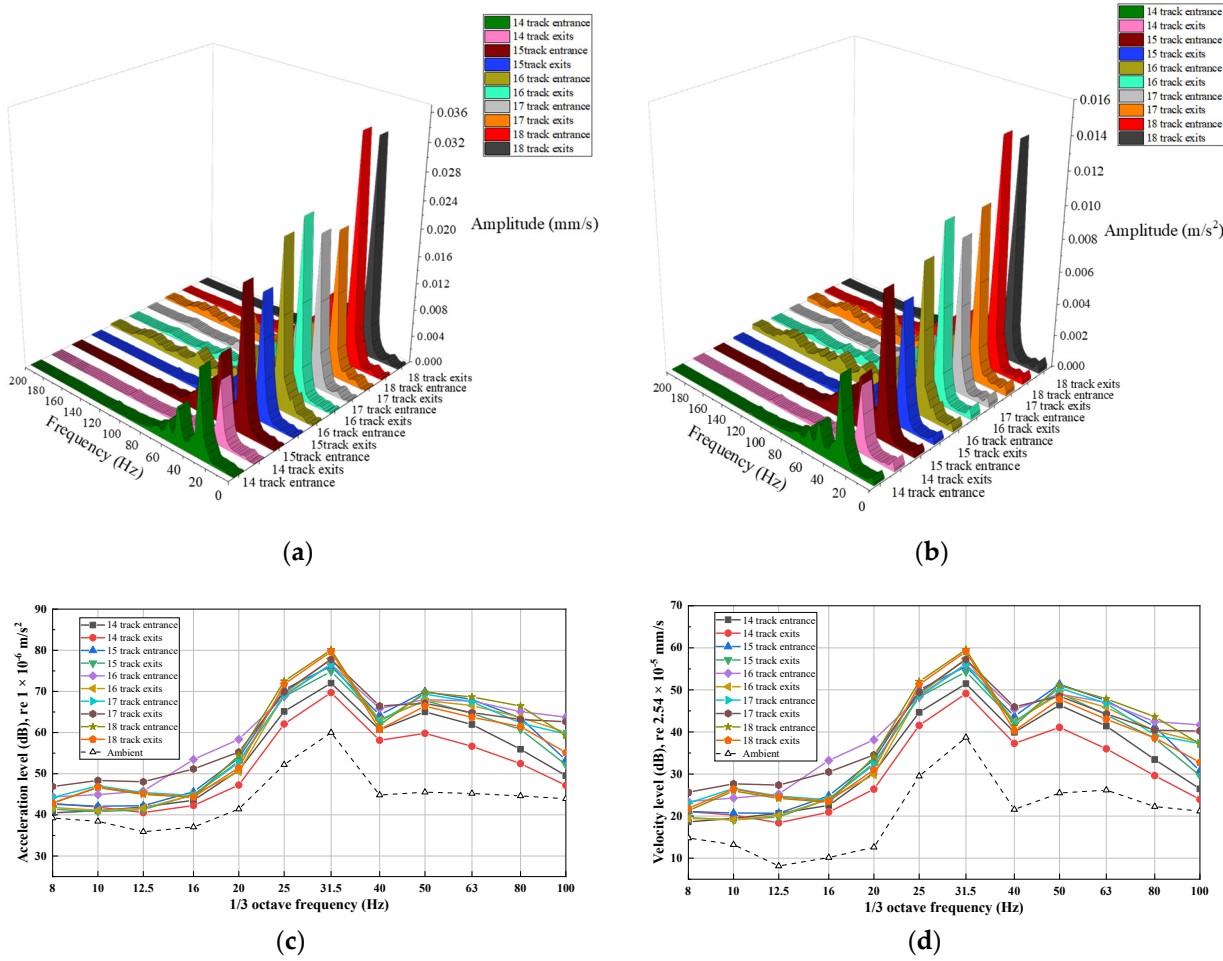

**Figure 8.** (**a**) Results of the building vibration acceleration caused by a train passing over different tracks in the frequency domain; (**b**) results of the building vibration velocity caused by a train passing over different tracks in the frequency domain; (**c**) results of the building vibration acceleration over one-third octave caused by a train passing over different tracks; (**d**) results of building vibration velocity over one-third octave caused by a train passing over different tracks.

### 3.2. Floor Effect on Vibration Response

This test was conducted uniformly on the same floors as the previous test (i.e., third, fifth, seventh, ninth, and 11th floors). The first floor was temporarily unavailable for testing due to construction problems, and the 11th floor was the top floor of the building.

The analysis was conducted for the entrance of track 15 and the exit of track 16 to determine the vibration responses at the entrance and exit of the operation depot on different floors. The acceleration was analyzed in the time and frequency domains in the bedroom on each of the test floors, to illustrate the transmission pattern of vibrations between floors.

The time–domain analysis results of the vibration response from the third to the 11th floors showed that the vibration intensity first decreased, then increased, and finally decreased; the vibration intensity was the largest on the upper-middle floor. The vibration attenuation between the floors showed a zigzag trend. Therefore, when assessing the vibration impact on a building, the lowest floor should not be selected for measurement and evaluation. The frequency-domain analysis results of the vibration response on the different floors showed that the main frequency band of vibrations was 25–80 Hz. The peak frequency of the third, fifth, ninth, and 11th floors was 63 Hz, while the peak frequency of the seventh floor was 50 Hz. The results in Figure 8 show that the vibration peak frequency of the drawing room was 31.5 Hz; meanwhile, the vibration peak frequencies

of the bedroom were 63 Hz and 50 Hz, as shown in Figures 9 and 10. In the subsequent analysis of the vibration effect on occupant comfort in the over-track buildings, the third and seventh floors were considered.

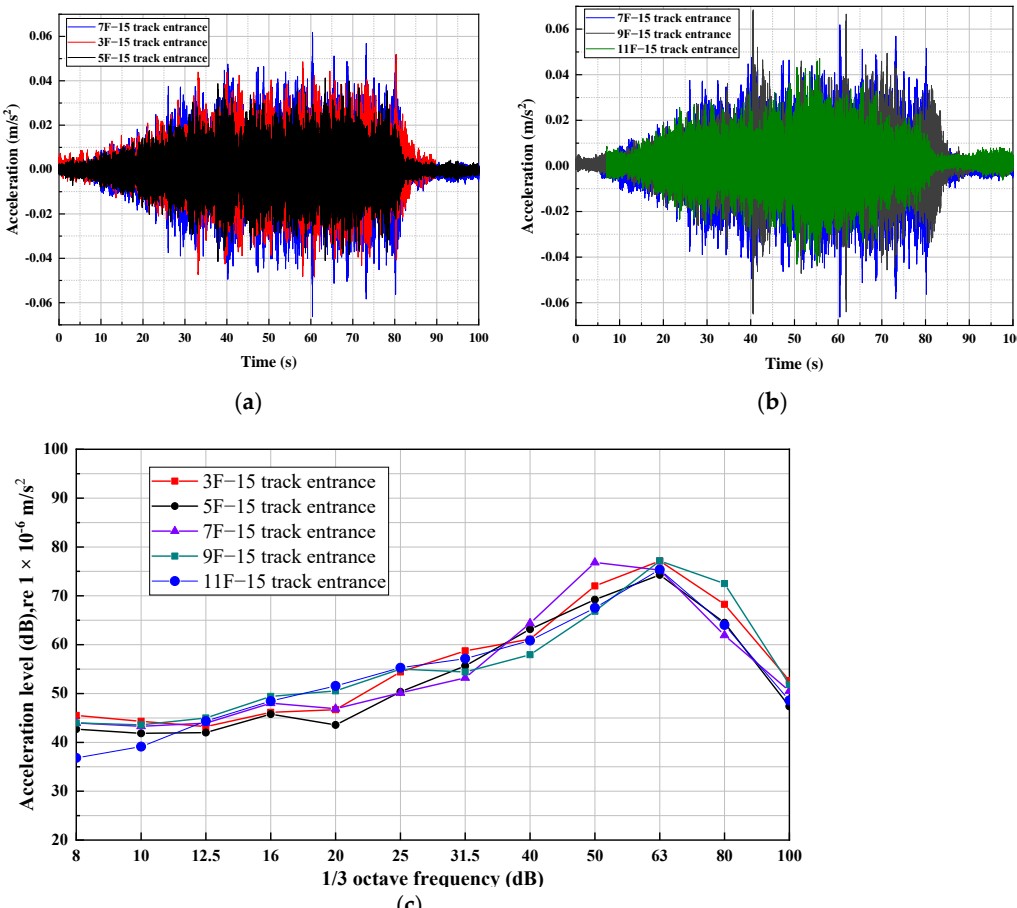

**Figure 9.** Analysis results of the vibration effect on human comfort on different floors in the time and frequency domains when the train enters track 15. (**a**) Time domain diagrams of measurement points on the 3rd, 5th and 7th floors; (**b**) Time domain diagrams of measurement points on the 7th, 9th and 11th floors; (**c**) One-third octave frequency diagrams of measurement points on the 3rd, 5th, 7th, 9th, and 11th floors.

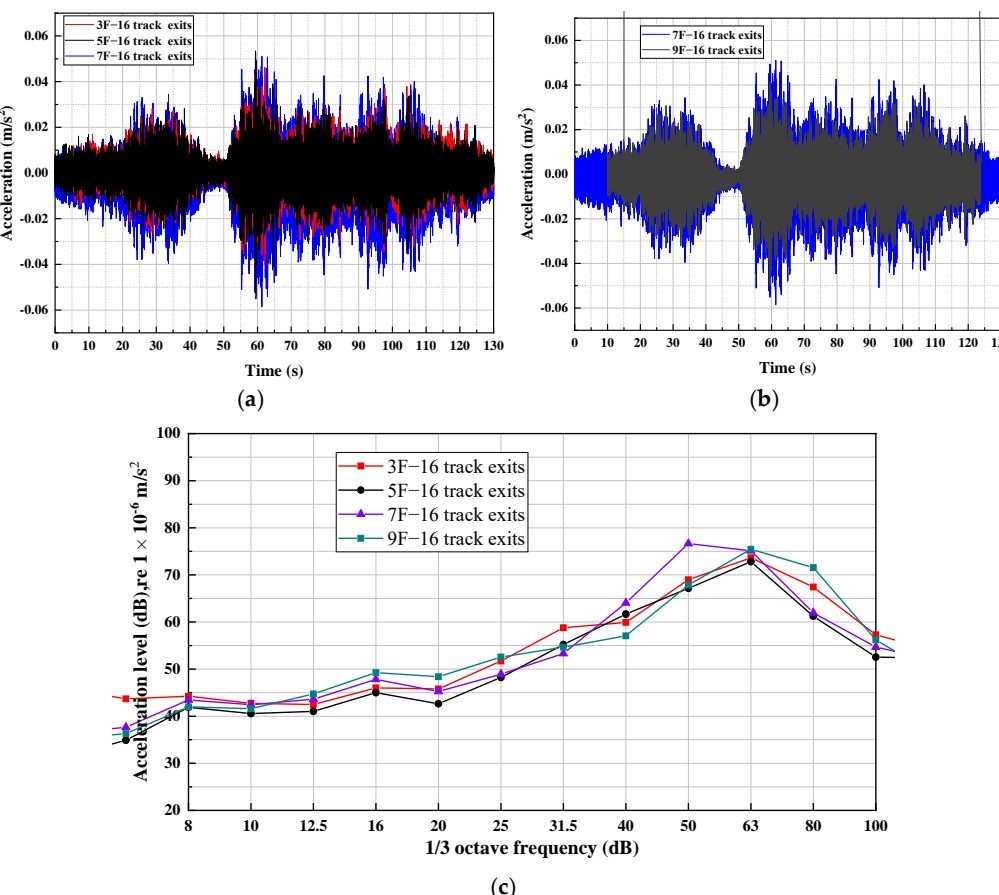

**Figure 10.** Analysis results of the vibration effect on human comfort on different floors in the time and frequency domains when the train exits track 16. (**a**) Time domain diagrams of measurement points on the 3rd, 5th and 7th floors; (**b**) Time domain diagrams of measurement points on the 7th and 9th floors; (**c**) One-third octave frequency diagrams of measurement points on the 3rd, 5th, 7th, and 9th floors.

## 4. Evaluation of Vibration Impact on Human Comfort

When evaluating the impact of vibrations on human comfort, it is usually referred to as the vibration standard evaluation. However, the vibration limit values defined by the standards around the world do not ensure that 100% of the population is not disturbed, the evaluation of vibration results cannot reflect the continuity of human subjective feelings, and the vibration standards cannot provide an evaluation of large complex systems. The analysis based on the annoyance rate could be beneficial to the evaluation of quantitative vibration comfort. In an environment with vibrations, the percentage of people who experience annoyance at the same vibration intensity in the total number of people in the environment represents the structural vibration annoyance rate. The annoyance rate caused by the vibrations of the over-track building is studied by using the two evaluation systems of acceleration and velocity, so as to summarize the influence of the vibrations of the over-track building on human comfort.

### 4.1. Human Comfort Analysis with Vibration Velocity as Evaluation Index

The effects of vibrations on human comfort have been extensively investigated and analyzed in many countries. The research results of developed countries (e.g., USA, Norway) on human comfort were selected in this study to evaluate human comfort in over-track buildings. It should be noted that different countries use different evaluation indicators and weighting methods, for example, the USA and Norway have adopted velocity indicators.

4.1.1. Annoyance Rate Analysis Based on Vibration Dosage-Response Curve Released by USA

The FTA (Federal Transit Administration) and FRA (Federal Railroad Administration) guidelines state that the human body responds to an average vibration amplitude. Because the net average of a vibration signal is zero, the root mean square (rms) amplitude is used to describe the "smoothed" vibration amplitude. Decibel notation is commonly used for vibration assessment ($L_v$ dB).

According to the FTA and FRA guidelines [29,30], the peak velocity level of the building structure, $L_v$, is calculated as follows:

$$L_v = 20 \log_{10} \left[ \frac{v}{v_{\text{ref}}} \right] \tag{1}$$

where $v$ is the rms velocity.

The reference velocity $v_{\text{ref}}$ is calculated as follows:

$$v_{\text{ref}} = 2.54 \times 10^{-8} \, \text{m/s} \tag{2}$$

The vibration data of the drawing room and bedroom on the third and seventh floors were analyzed according to the above formula, and the results are shown in Table 4. The Transportation Cooperative Research Program (TCRP), sponsored by the FTA in cooperation with Transit Development Corporation and administered by the Transportation Research Board of the National Academy of Sciences, has studied the annoyance rate due to vibrations, and the annoyance rate curve is shown in Figure 11, where 72 dB is the limit for frequent events in residential areas.

For a normal distribution, the probability that a train would exceed the mean +2σ level is 5%, so this level would correspond to the "loudest" trains that passed by the site, and, hence, is the more appropriate measure if it is believed that people are more likely to be disturbed by the loudest trains in the fleet, rather than the fleet-average train. The results obtained, using the method based on the mean plus two times the standard deviation in the American Social Vibration Surveys-Annoyance, are presented in Table 5.

The results indicated that the vibrations in the seventh-floor drawing room would have a 9% probability of making the residents feel highly annoyed, and a 17.5% probability of making them feel moderately or highly annoyed; the vibrations in the seventh-floor bedroom would have a 6.5% probability of making residents feel highly annoyed, and a 11.5% probability of making them feel moderately or highly annoyed. Further, the vibrations in the third-floor drawing room would have a 6.2% probability of making residents feel highly annoyed and a 11.2% probability of making them feel moderately or highly annoyed; the vibrations in the third-floor bedroom would have a 3.5% probability of making residents feel highly annoyed and a 6.5% probability of making them feel moderately or highly annoyed. Figure 12 compares the train-induced vibration levels in the over-track building with the vibration limits categorized under frequent events in the US surveys. Only the bedroom vibrations on the third floor did not exceed the limits, with the maximum vibration level occurring in the living room on the seventh floor, which exceeded the limit by 6.93 dB.

**Table 4.** Evaluation of vibration levels at the measurement point according to the US vibration standards.

| Track | Drawing Room-3F $L_v$ (dB) | Bed Room-3F $L_v$ (dB) | Drawing Room-7F $L_v$ (dB) | Bed Room-7F $L_v$ (dB) |
|---|---|---|---|---|
| 14-track entrance | 57.81 | 58.29 | 60.02 | 61.03 |
| 14-track exits | 57.89 | 55.10 | 59.35 | 59.16 |
| 15-track entrance | 67.01 | 62.92 | 65.20 | 66.62 |
| 15-track exits | 64.22 | 63.27 | 63.20 | 65.10 |
| 16-track entrance | 66.60 | 64.04 | 77.17 | 70.68 |
| 16-track exits | 66.42 | 65.10 | 77.26 | 67.27 |
| 17-track entrance | 66.90 | 63.32 | 65.71 | 70.77 |
| 17-track exits | 70.34 | 64.34 | 67.57 | 69.60 |
| 18-track entrance | 68.61 | 62.14 | 67.21 | 68.08 |
| 18-track exits | 66.78 | 59.09 | 69.66 | 67.88 |

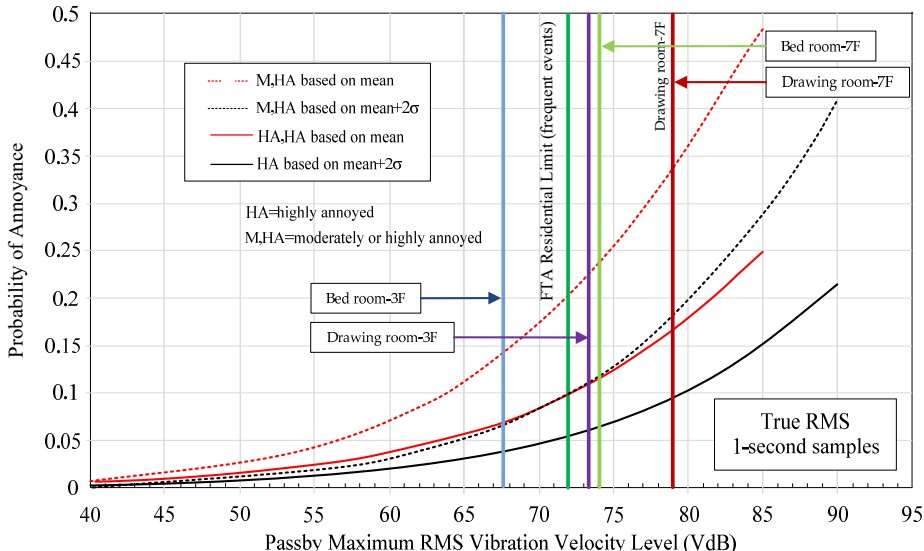

**Figure 11.** Results of the American Social Vibration Surveys-Annoyance [30].

**Table 5.** Annoyance assessment results obtained by the (Mean + 2σ) method.

| Floors and Rooms | Velocity Level Mean + 2σ (dB) | Probability of Highly Annoyed | Probability of Moderately or Highly Annoyed |
|---|---|---|---|
| Drawing room-3F | 73.24 | 0.062 | 0.112 |
| Bed room-3F | 67.85 | 0.035 | 0.065 |
| Drawing room-7F | 78.93 | 0.090 | 0.175 |
| Bed room-7F | 73.99 | 0.065 | 0.115 |

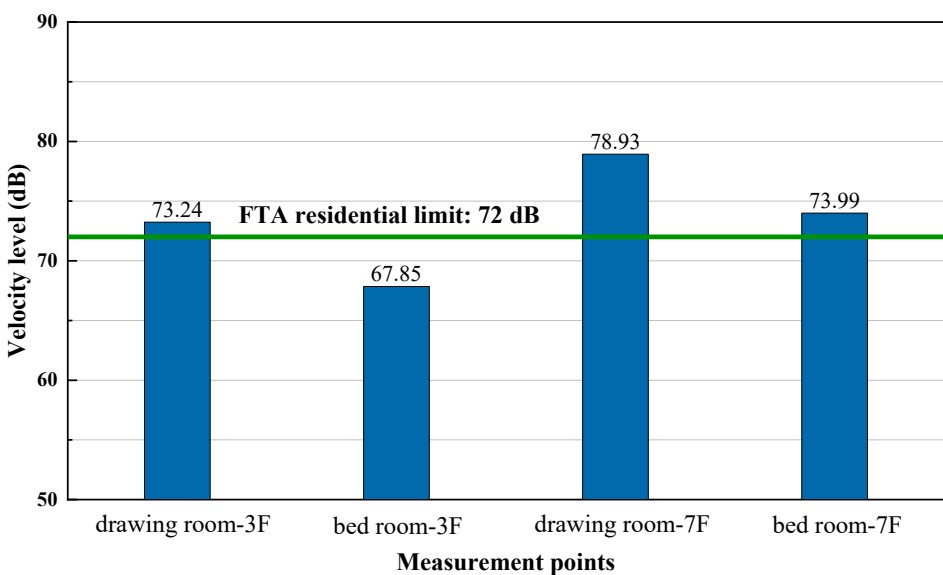

**Figure 12.** Comparison of vibration of the over-track building and the US standard.

4.1.2. Annoyance Rate Analysis Based on Vibration Exposure–Response Curve Released by Norway

The Norwegian standard (NS 8176:2005) suggests using the statistical maximum weighted acceleration or velocity level ($a_{w,95}$ or $v_{w,95}$) when assessing the vibration effect on human comfort [31]. These indicators are calculated from the 1-s rms averages of $w_m$-weighted acceleration or velocity signals. In this study, the velocity index was selected for evaluation.

The $v_{w,95}$ descriptor was calculated as follows:

$$v_{w,95} = \overline{v_{w,\max}} + 1.8\sigma_v \tag{3}$$

where $\overline{v_{w,\max}}$ is the average value of the maximum weighted speed of all trains passing during the evaluation period; and $\sigma_v$ is the standard deviation of the maximum weighted speed of all of the trains passing during the evaluation period.

The $\overline{v_{w,\max}}$ and $\sigma_v$ values are, respectively, calculated as follows:

$$\overline{v_{w,\max}} = \frac{\sum\limits_{j=1}^{N} v_{w,\max,j}}{N} \tag{4}$$

$$\sigma = \sqrt{\frac{1}{N-1}\sum_{j=1}^{N}\left(v_{w,\max,j} - \overline{v_{w,\max}}\right)^2} \tag{5}$$

where $v_{w,\max}$ is the maximum 1-s average weighted speed of a single train passing during the evaluation period; and $N$ is the total number of trains passing during the evaluation period.

The value of $v_{w,\max}$ can be calculated using various standard arithmetic methods, but this study adopts the $w_m$-frequency weighting method. The exposure effect curves for the rest and daily activity periods were evaluated according to the curves defined by the standard. As shown in Figure 13, there were between 10 and 15 reports of disturbances during the rest and sleep periods, respectively, and $v_{w,95}$ was about 0.1 mm/s. This test was conducted for the trains running in and out of the depot on tracks 14–18, for a total of 10 trains, to simulate the normal operation of the train. Finally, a standard assessment was performed for all of the 10 trains.

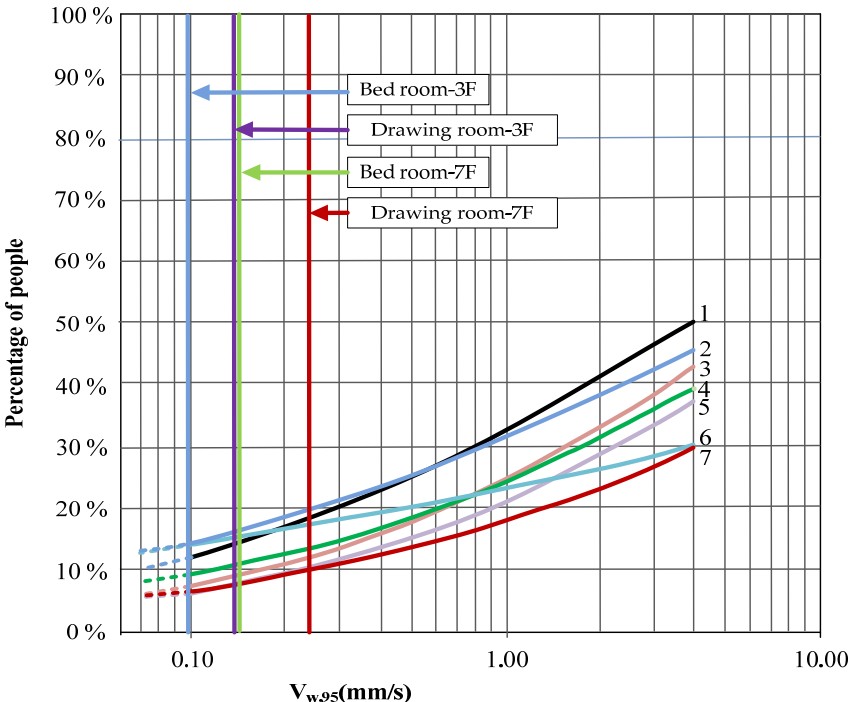

**Figure 13.** Percentage of people who were disturbed during rest or during daily activities by the vibrations in dwellings. The results are plotted against the calculated statistical maximum value for the weighted velocity, $v_{w,95}$, expressed in mm/s [17].

Where key:

1.  Disturbance when watching TV or listening to the radio;
2.  Disturbance during the rest period;
3.  Waking up too early;
4.  Waking up during the night;
5.  Disturbance during telephone usage;
6.  Disturbance during conversations;
7.  Difficulty in falling asleep.

Table 6 shows the results of $v_{w,max}$ and $v_{w,95}$ on the third floor when the train passed through tracks 14–18. The $v_{w,95}$ value in the drawing room was 0.14; considering the function of this room, the vibrations affected 14% of people watching TV or engaging in other sources of entertainment, 8% of people talking on the phone, and 15% of people talking to each other. The $v_{w,95}$ value in the bedroom was 0.09; considering the function of this room, the vibrations affected 12% of people who were resting, and 7% of people reported difficulty in sleeping.

Table 7 shows the results of $v_{w,max}$ and $v_{w,95}$ on the seventh floor when the train passed through tracks 14–18. The $v_{w,95}$ value in the drawing room was 0.23; considering the function of this room, the vibrations affected 19% of people watching TV or engaging in other sources of entertainment, 11% of people talking on the phone, and 18% of people talking to each other. The $v_{w,95}$ value in the bedroom was 0.15; considering the function of this room, the vibrations affected 17% of people who were resting, and 9% of people experienced certain difficulties in sleeping. Figure 14 compares the train-induced vibration levels in the over-track building with the Norwegian standard vibration limits, where vibrations above Class B causes a certain level of vibration disturbance to residents; the living room and bedroom on the seventh floor exceed the limits of Class B.

**Table 6.** Evaluation indicators for the third-floor rooms.

| Table. | Drawing Room $v_{w,max}$ (mm/s) | Bed Room $v_{w,max}$ (mm/s) | Drawing Room $v_{w,max}$ (mm/s) | Bed Room $v_{w,max}$ (mm/s) |
|---|---|---|---|---|
| 14-track entrance | 0.0385 | 0.03777 | | |
| 14- track exits | 0.0319 | 0.03178 | | |
| 15-track entrance | 0.0777 | 0.064694 | | |
| 15- track exits | 0.06817 | 0.057754 | | |
| 16-track entrance | 0.10331 | 0.087049 | | |
| 16- track exits | 0.0839 | 0.065383 | 0.14 | 0.09 |
| 17-track entrance | 0.0873 | 0.067918 | | |
| 17- track exits | 0.1266 | 0.079552 | | |
| 18-track entrance | 0.11804 | 0.056475 | | |
| 18- track exits | 0.11611 | 0.040929 | | |

**Table 7.** Evaluation indicators for the seventh-floor rooms.

| Track | Drawing Room $v_{w,max}$ (mm/s) | Bed Room $v_{w,max}$ (mm/s) | Drawing Room $v_{w,95}$ (mm/s) | Bed Room $v_{w,95}$ (mm/s) |
|---|---|---|---|---|
| 14-track entrance | 0.0461 | 0.0404 | | |
| 14-track exits | 0.0411 | 0.0374 | | |
| 15-track entrance | 0.0862 | 0.0831 | | |
| 15-track exits | 0.0760 | 0.0695 | | |
| 16-track entrance | 0.3093 | 0.1378 | | |
| 16-track exits | 0.1076 | 0.0929 | 0.23 | 0.15 |
| 17-track entrance | 0.0939 | 0.1263 | | |
| 17-track exits | 0.0939 | 0.1326 | | |
| 18-track entrance | 0.1013 | 0.0907 | | |
| 18-track exits | 0.1095 | 0.0940 | | |

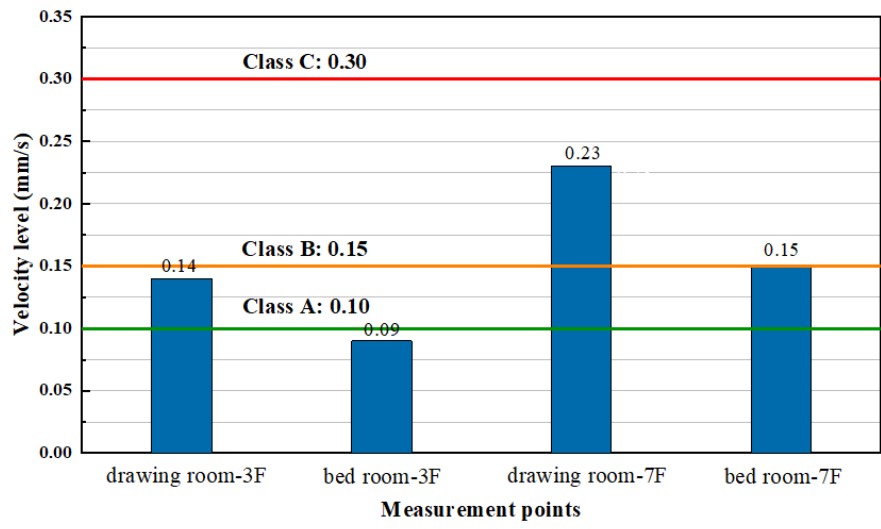

**Figure 14.** Comparison of vibration of the over-track building and Norway standard. Class A, normally not be expected to notice vibration; Class B, can be expected to be disturbed by vibration to some extent; Class C, about 15% of the affected persons in the dwellings can be expected to be disturbed by vibration.

*4.2. Human Comfort Analysis with Vibration Acceleration as Evaluation Index*

4.2.1. Annoyance Rate Analysis Based on Vibration Exposure–Response Curve Released by Europe

The EU FP7 project, CargoVibes, conducted a meta-analysis of a total of 4490 existing samples of surveys to investigate the effect of vibrations on the population, providing the

exposure–response relationship curves for Germany, Norway, Japan, the United States, Canada, the United Kingdom, Sweden, the Netherlands, and Poland [16]. The curves are plotted for three different vibration exposure descriptors to provide a reasonable estimate of annoyance from the vibration exposure measurement, according to the major standards. Figure 15 shows the annoyance rate plotted for the *rms* indicator. The impact of the vibrations generated by train operation on the comfort of occupants in the building was evaluated considering the peak-hour operation at night and early morning and a single-train operation for daytime shunting conditions. The three vibration exposure descriptors are as follows:

(1)  $V_{dir,max}$: Maximum $w_k$-weighted fast exponentially filtered rms velocity over the entire evaluation period;
(2)  *rms*: $w_k$-weighted rms acceleration over the entire evaluation period;
(3)  *VDV*: $w_k$-weighted vibration intensity over the entire evaluation period.

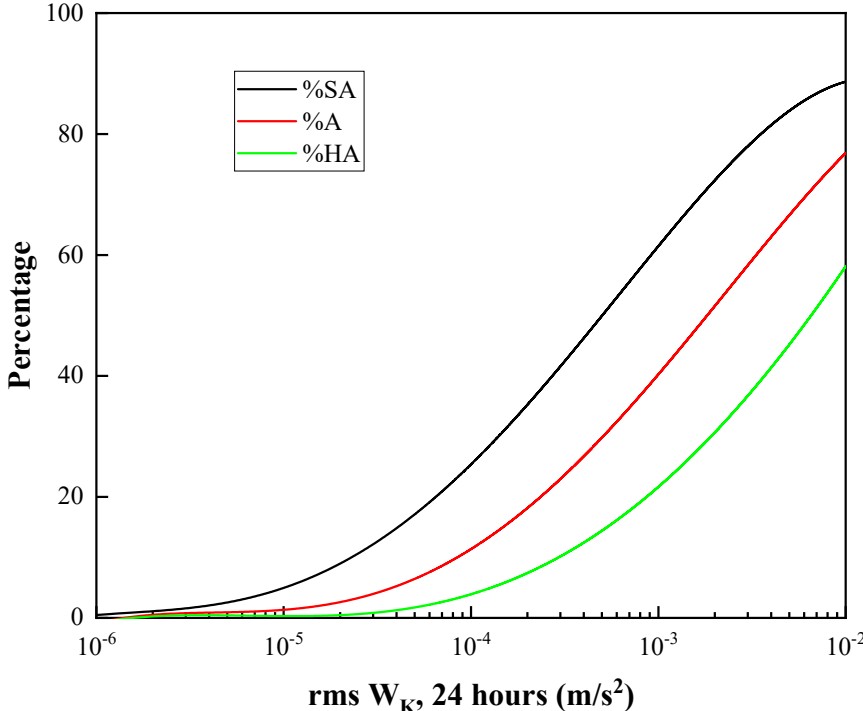

**Figure 15.** Annoyance rate for the *rms* ($w_k$-weighted root-mean-square acceleration over the entire evaluation period) indicator.

For the human comfort evaluation in the over-track building using *rms*, the annoyance rate is calculated as follows:

$$\%SA_{rms} = -1.806X^4 - 3.198X^3 + 11.812X^2 + 35.059X + 25.390 \tag{6}$$

$$\%A_{rms} = -1.648X^4 - 0.013X^3 + 13.826X^2 + 22.510X + 11.380 \tag{7}$$

$$\%HA_{rms} = -0.527X^4 + 2.089X^3 + 9.850X^2 + 10.785X + 3.910 \tag{8}$$

where $SA_{rms}$ denotes slight annoyance; $A_{rms}$ denotes annoyance; $HA_{rms}$ denotes severe annoyance; and $X$ is related to the rms value of the acceleration.

The $X$ descriptor is calculated as follows:

$$X = \frac{\log_{10}(\text{rms}) + 4}{1.1564} \tag{9}$$

It is important to note that the above equations must not be used when the value of rms is out of range of $(0.001 \times 10^{-3}, 10 \times 10^{-3})$ m/s$^2$.

(1)　The annoyance rate during peak hours of train operation

Next, the annoyance rates were analyzed in the rooms on the third and seventh floors, and the results showed that the third-floor drawing room had a high annoyance rate during the peak hours, with the highest HA of 8.27%, which severely affected the comfort of the residents. The bedroom on the third floor also had a high annoyance rate during the peak hours, with the highest HA of 8.49%, which occurred between 5:30 a.m. and 6:30 a.m. The results are shown in Table 8. However, this period comprises the residents' sleep time; thus, any annoyance in this period can affect the residents' sleep.

**Table 8.** Vibration annoyance rate of the third-floor test room during peak hours of train operation.

| Single Peak Hour | Test Room-3F | %HA | %A | %SA |
|---|---|---|---|---|
| 10:30 p.m. to 11:30 p.m. | Drawing room | 4.24 | 12.07 | 26.46 |
| 11:30 p.m. to 12:30 p.m. | Drawing room | 6.30 | 16.09 | 32.36 |
| 4:30 a.m. to 5:30 a.m. | Drawing room | 5.18 | 13.95 | 29.28 |
| 5:30 a.m. to 6:30 a.m. | Drawing room | 8.27 | 19.69 | 37.31 |
| 10:30 p.m. to 11:30 p.m. | Bed room | 8.71 | 20.47 | 38.34 |
| 10:30 p.m. to 12:30 p.m. | Bed room | 7.91 | 19.05 | 36.45 |
| 4:30 a.m. to 5:30 a.m. | Bed room | 7.96 | 19.14 | 36.57 |
| 5:30 a.m. to 6:30 a.m. | Bed room | 8.49 | 20.08 | 37.82 |

The annoyance rate measured in the bedroom on the seventh floor was smaller compared to that on the third floor; thus, did not have a significant impact on the comfort of the residents, as shown in Table 9.

**Table 9.** Vibration annoyance rate of the seventh-floor test room during peak hours of train operation.

| Single Peak Hour | Test Room-7F | %HA | %A | %SA |
|---|---|---|---|---|
| 10:30 p.m. to 11:30 p.m. | Bed room | 0.59 | 3.13 | 10.28 |
| 10:30 p.m. to 12:30 p.m. | Bed room | 0.69 | 3.49 | 11.13 |
| 4:30 a.m. to 5:30 a.m. | Bed room | 0.58 | 3.09 | 10.18 |
| 5:30 a.m. to 6:30 a.m. | Bed room | 0.84 | 3.98 | 12.22 |

The analysis results of the peak hours showed that the annoyance rate during peak hours was higher than that during other periods of the day. Since the trains stop operating at night and depart again early in the morning, and the late evening and early morning are sleeping times, in these times, the vibration annoyance rate in the bedroom has a high impact on people's comfort, from the point of view of the room's function.

(2)　The annoyance rate in a single-train operation

For a more detailed assessment of the impact of the different tracks on building vibrations and thus human comfort, calculations were performed for all of the tracks, and the results are presented in Tables 10 and 11.

The calculation was performed with a 5% annoyance rate limit to ensure that the comfort of 95% of people is relatively guaranteed. In the drawing room on the third floor, the annoyance rate exceeded the limit for all of the tracks. The train ran 10 times a night, and the vibrations generated in 8 of 10 cases caused discomfort to the residents. It should be noted that even the vibrations from only one operation caused discomfort to the residents in the bedrooms on both the third and the seventh floors.

**Table 10.** Vibration annoyance rate of the third-floor test room during a single-train operation.

| A Single Car | Test Room-3F | %HA | %A | %SA |
|---|---|---|---|---|
| 14-track entrance | Drawing room | 4.70 | 12.99 | 27.85 |
| 14-track exits | Drawing room | 2.80 | 8.99 | 21.55 |
| 15-track entrance | Drawing room | 6.98 | 17.37 | 34.15 |
| 15-track exits | Drawing room | 5.00 | 13.59 | 28.75 |
| 16-track entrance | Drawing room | 5.26 | 14.10 | 29.49 |
| 16-track exits | Drawing room | 6.56 | 16.59 | 33.06 |
| 17-track entrance | Drawing room | 6.03 | 15.58 | 31.64 |
| 17-track exits | Drawing room | 5.91 | 15.36 | 31.33 |
| 18-track entrance | Drawing room | 8.31 | 19.77 | 37.41 |
| 18-track exits | Drawing room | 7.59 | 18.48 | 35.68 |
| 14-track entrance | Bed room | 4.88 | 13.36 | 28.41 |
| 14-track exits | Bed room | 2.58 | 8.48 | 20.69 |
| 15-track entrance | Bed room | 5.24 | 14.07 | 29.45 |
| 15-track exits | Bed room | 3.92 | 11.39 | 25.41 |
| 16-track entrance | Bed room | 3.68 | 10.90 | 24.64 |
| 16-track exits | Bed room | 3.72 | 10.98 | 24.76 |
| 17-track entrance | Bed room | 3.44 | 10.39 | 23.83 |
| 17-track exits | Bed room | 3.21 | 9.90 | 23.04 |
| 18-track entrance | Bed room | 4.36 | 12.31 | 26.82 |
| 18-track exits | Bed room | 3.33 | 10.15 | 23.45 |

**Table 11.** Vibration annoyance rate of the seventh-floor test room during a single-train operation.

| A Single Car | Test Room-7F | %HA | %A | %SA |
|---|---|---|---|---|
| 14-track entrance | Bed room | 2.38 | 8.02 | 19.90 |
| 14-track exits | Bed room | 1.20 | 5.04 | 14.41 |
| 15-track entrance | Bed room | 3.23 | 9.93 | 23.09 |
| 15-track exits | Bed room | 2.44 | 8.16 | 20.15 |
| 16-track entrance | Bed room | 5.92 | 15.37 | 31.34 |
| 16-track exits | Bed room | 3.60 | 10.72 | 24.36 |
| 17-track entrance | Bed room | 4.15 | 11.87 | 26.16 |
| 17-track exits | Bed room | 4.72 | 13.03 | 27.91 |
| 18-track entrance | Bed room | 4.36 | 12.31 | 26.83 |
| 18-track exits | Bed room | 2.58 | 8.49 | 20.70 |

### 4.2.2. Annoyance Rate Analysis Using the Set-Value Statistical Method and Psychological Annoyance Rate

Due to the vagueness and randomness of the subjective vibration response judgment, the set-value statistical method and psychological annoyance rate were used to evaluate human comfort. The structural vibration annoyance rate calculation method, introduced by Tu et al. [32] and Song [33], was adopted.

For the case of discrete distribution, the annoyance rate is calculated as follows:

$$A(a_{wi}) = \frac{\sum_{j=1}^{m} v_j n_{ij}}{\sum_{j=1}^{m} n_{ij}} = \sum_{j=1}^{m} v_j p(i,j) \tag{10}$$

where $A(a_{wi})$ is the annoyance rate at the $i$th vibration intensity $a_{wi}$; $n_{ij}$ is the number of people with the $j$th subjective response at the $i$th vibration intensity; $v_j$ is the conceptual affiliation of the "unacceptable" category of the $j$th subjective response; $\sum_{j=1}^{m} n_{ij}$ is the total number of people reporting "unacceptable" vibration intensity; and $p(i, j)$ reflects the variability in the annoyance degree among people.

The value of $v_j$ is calculated as follows:

$$v_j = \frac{j-1}{m-1} \tag{11}$$

where $m$ is the number of levels of the subjective response of people, and usually, $m$ is set to 5 or 11; when $m = 5$, the levels of the subjective response are: "no vibration"; "light vibration"; "moderate vibration"; "strong vibration"; and "unbearable."

For the case of continuous distribution, since there is a variability in the human perception of vibrations, the variability in people's responses can be described by a log-normal distribution [34].

Considering the distribution characteristics of ambiguity and randomness, the annoyance rate under continuous distribution at the vibration acceleration $x$ is given by:

$$A(x) = \int_{u_{\min}}^{\infty} \frac{1}{\sqrt{2\pi}u\sigma_{\ln}} \exp\left(\frac{-\left(\ln(u/x) - 0.5\sigma_{\ln}^2\right)^2}{2\sigma_{\ln}^2}\right) v(u)\,du \tag{12}$$

where $x$ is the expected value of $u$; and $v(u)$ is the vibration intensity fuzzy affiliation function.

Further, the value of $\sigma_{\ln}$ is calculated as follows:

$$\sigma_{\ln} = \sqrt{\ln(1 + \delta^2)} \tag{13}$$

and $v(u)$ is calculated as follows:

$$v(u) = \begin{cases} 0 & u < u_{\min}, \\ a\ln(u) + b & u_{\min} \le u \le u_{\max} \\ 1 & u > u_{\max}, \end{cases} \tag{14}$$

where $\delta$ is the coefficient of variation of $u$, and it is usually set to 0.3 [33,35]; $u_{\min}$ is the upper limit of the vibration intensity that a human defines as "not felt"; $u_{\max}$ is the lower limit of the vibration intensity that a human being senses as "unbearable". Based on the experiments, the values of $u_{\min}$ and $u_{\max}$ are 0.05 m/s$^2$ and 1.5 m/s$^2$, respectively; $a$ and $b$ are coefficients to be determined, and they are calculated as follows:

$$\begin{cases} a\ln(u_{\min}) + b = 0, \\ a\ln(u_{\max}) + b = 1, \end{cases} \tag{15}$$

The annoyance rate can be regarded as a resistance $R$ in the reliability analysis, and its distribution function can be expressed by a log-normal distribution function. Therefore, approximating the annoyance rate curve before the analysis of the annoyance rate could be a good solution to obtain objective results, and the calculation result approximation does not cause significant deviations [33].

The function $A(x)$ is expressed as follows:

$$A(x) \approx CDF_{\log norm}(x, \mu_{\ln x}, \sigma_{\ln x}) \tag{16}$$

where, in the vertical direction of the train-induced vibrations, $\mu_{\ln x}$ has a value of $-4.247$, and $\sigma_{\ln x}$ equals 0.473.

By using the calibration method, an engineering acceptable design level of vibration comfort corresponding to an allowable annoyance rate of 7% is obtained [33]. The vibration data of the third and seventh floors of the over-track buildings were analyzed and evaluated, using Equation (16).

For a more detailed assessment of the impact of the different tracks on the building vibrations, calculations were performed for all of the tracks, as shown in Tables 12 and 13. In the drawing room on the third floor, the annoyance rate for all of the other tracks, except

for track 14, exceeded the limit. The train ran 10 times a night, and the vibrations generated in 7 of 10 cases caused discomfort to the residents. For occupants living in the bedrooms of the third and seventh floors, the annoyance rate caused by vibrations did not exceed the 7% limit; thus, the vibrations did not have a noticeable effect on the comfort of the residents in these bedrooms.

**Table 12.** Vibration annoyance rate of the third-floor test room during a single-train operation based on annoyance rate model.

| A Single Car | Test Room-3F | Annoyance Rate |
| --- | --- | --- |
| 14-track entrance | Drawing room | 0.086% |
| 14-track exits | Drawing room | 0.14% |
| 15-track entrance | Drawing room | 15.89% |
| 15-track exits | Drawing room | 6.16% |
| 16-track entrance | Drawing room | 19.82% |
| 16-track exits | Drawing room | 17.41% |
| 17-track entrance | Drawing room | 21.75% |
| 17-track exits | Drawing room | 51.71% |
| 18-track entrance | Drawing room | 37.2% |
| 18-track exits | Drawing room | 21.19% |
| 14-track entrance | Bed room | 0.00059% |
| 14-track exits | Bed room | 0.0003% |
| 15-track entrance | Bed room | 0.16% |
| 15-track exits | Bed room | 0.088% |
| 16-track entrance | Bed room | 0.29% |
| 16-track exits | Bed room | 0.35% |
| 17-track entrance | Bed room | 0.11% |
| 17-track exits | Bed room | 0.2% |
| 18-track entrance | Bed room | 0.072% |
| 18-track exits | Bed room | 0.00018% |

**Table 13.** Vibration annoyance rate of the seventh-floor test room during a single-train operation based on annoyance rate model.

| A Single Car | Test Room-7F | Annoyance Rate |
| --- | --- | --- |
| 14-track entrance | Bed room | 0.001% |
| 14-track exits | Bed room | 0.000% |
| 15-track entrance | Bed room | 0.0159% |
| 15-track exits | Bed room | 0.0089% |
| 16-track entrance | Bed room | 0.1479% |
| 16-track exits | Bed room | 0.0207% |
| 17-track entrance | Bed room | 0.1713% |
| 17-track exits | Bed room | 0.0898% |
| 18-track entrance | Bed room | 0.0508% |
| 18-track exits | Bed room | 0.0498% |

## 5. Findings and Discussion

Based on the above measurement campaign and human comfort evaluation, it was clear that the neighborhood residents were indeed suffering from the railway-induced vibrations under certain circumstances. However, as the human comfort would be related to the vibration level and period of period, the serviceability and particularity of the TOD developed depot still need to be fully discussed.

### 5.1. The Particularity of Railway Vibration at Tod Developed Depot

Firstly, a metro depot is a facility where trains are regularly parked for maintenance, testing, and storage. Therefore, there were rush hours when the metro trains started going out very early in the morning and coming back very late at night. The main uncomfortable

influence due to railway vibrations would be that causing residents difficulty in sleeping at night or waking early in the morning.

Secondly, based on the measurement results, the highest vibrations happened when the train was running though the track immediately under the particular bedroom. Generally, during the rush hours, the resident may have to suffer one or two episodes of high-level vibration shock and four to six episodes of sensible vibrations.

Thirdly, the level of the vibrations was highly related to train speed, as limited train speed would obviously reduce the vibrations.

Besides that, the structure of the building plays an important role in the vibration transit. The measurement results showed that when the vibration is transmitted to the upper floors, the vibration intensity first decreases, then increases, and then again decreases; so, the vibrations are the largest on the upper-middle floors. Thus, when assessing the vibration impact, not only the lowest floors should be selected for measurement and evaluation, but also the upper-middle floors.

*5.2. Human Comfort Analysis with Vibration Velocity as Evaluation Index*

There are two types of human response index associated with vibration velocity in the available national standards. The US standards provide a single-figure exposure descriptor related to the energy equivalent *rms* velocity value and vibration velocity level, considering frequency weightings, which in the presented case study, means that the residents in the seventh floor drawing room will be most disturbed, with a 9% probability of being highly annoyed and a 17.5% probability of being moderately or highly annoyed. In the Norwegian standards, detailed human comfort responses to vibration velocity are provided, including having sleeping and living activities disturbed, which considers a frequency weighting index $w_m$ in the range of 0.5 to 160 Hz. In the presented case study, the most serious impact occurs in the bedroom on the seventh floor; that particular vibration affected 17% of the people who were resting and 9% of people who had difficulties in sleeping. It is suggested that the sleep disturbed index should be more significant in the TOD developed depot, as the rush hours happen very late in the night and very early in the morning.

*5.3. Analysis of Human Comfort Using Train Acceleration as Evaluation Index*

To assess human comfort due to vibrations, the cumulative vibration acceleration values are more commonly used in some national standards as well. The acceleration indicators defined by the EU FP7 project, CargoVibes, and the annoyance rate based on the set-value statistical method and psychological annoyance rate are used for comfort assessment.

According to the EU FP7 project, CargoVibes, which weighted the root-mean-square acceleration for the entire evaluation period, using $w_k$ frequency weighting considering 1–80 Hz., in the study, the bedroom on the third floor reached a high annoyance rate during the peak hours, with an HA of 8.71%, occurring between 10:30 PM and 11:30 PM, which is the sleep time of the residents, and therefore affecting their sleep. The bedroom on the seventh floor has a lower annoyance rate, which did not significantly affect the comfort of the residents. During the daytime scheduled shunting, the annoyance rate was calculated with a 5% limit to ensure that the comfort of 95% of the people concerned was relatively guaranteed. In the drawing room on the third floor, the annoyance rate exceeded the limit for all of the tracks. According to the train timetable, the trains run 10 times during the nighttime, and the vibrations generated by the trains caused discomfort to the residents in 8 out of the 10 cases. Only one vibration would cause discomfort to the residents of the third and seventh floor bedrooms.

According to the annoyance rate based on the set-value statistical method and psychological annoyance rate, which uses $w_i$ frequency weighting considering 1–80 Hz., in the study, in the drawing room on the third floor, the annoyance rate for all of the tracks, except for track 14, exceeds the limit. The trains run 10 times at night, but in seven cases, the vibrations generated by the trains causes discomfort to the residents. In the bedrooms

on the third and seventh floors, the vibration limit is not reached; thus, the vibrations do not have a noticeable effect on human comfort.

Evaluating the acceleration effects in the over-track building, using the EU FP7 project, CargoVibes, and the annoyance rate based on the set-value statistical method and psychological annoyance rate, the vibrations are less annoying for people in the bedroom during the daytime scheduled shunting. However, at 10:30 to 11:30 p.m., the people in the third-floor bedrooms suffer from a high level of annoyance of 8.71%, which merits attention.

## 6. Conclusions

The present paper provided a case study to describe the effects of the metro railway-induced vibrations on human comfort at a particular TOD developed depot. In this paper, a measurement campaign was conducted at an operated metro depot, where the residential buildings were directly located on the cover structure of the metro train storage. Considering that there was not a proper standard to describe the human comfort subject to railway vibrations, several evaluation indexes were employed to analyze the structural serviceability in this paper. Some interesting findings could be found from this particular project, as below:

(1) It was clear that the neighborhood residents were indeed suffering from the railway-induced vibrations under certain circumstances. However, the results indicated that 90% of the occupants were not highly annoyed by the train-induced vibrations;

(2) The vibration events that happened at the TOD depot related to many factors, such as the train speed, building structure, and the track location that the trains were running on. It is possible to reduce the vibration effect by using a particular solution;

(3) At the particular situation of the metro depot, the main negative effect on human comfort was that the high level vibrations regularly happened in the rush hours, very early in the morning and very late at night, which would cause an interruption in sleep. Therefore, the evaluation index should consider more factors related to sleep difficulties and the awake threshold value;

(4) Based on the review of the current available standards, there are differences in terms of the single-figure or comprehensive indexes' descriptors, frequency weightings, measurement methods, and the guidelines' values for detailed impact. However, the current descriptors were insufficient to assess the effect of the vibrations on human comfort in such a particular situation, as it is difficult to derive exposure–response relationships or threshold values for impact on sleep and other living activities. Future studies should therefore focus on self-reported sleep difficulties and the impact on activities undertaken when awake from the residents living in TOD developed metro depots.

**Author Contributions:** Investigation, Y.L., P.Z., Q.W. and H.Z.; methodology, Y.L. and P.Z.; software, P.Z.; project administration, H.Z. and Y.L.; supervision, H.Z. and Y.L.; writing-original draft, P.Z. and H.Z.; writing-review and editing, H.Z. and Y.L. All authors have read and agreed to the published version of the manuscript.

**Funding:** This research was funded by the Beijing Natural Science Foundation, grant number 8202040, and the National Natural Science Foundation of China, grant number 12072208, and the Opening Foundation of State Key Laboratory of Shijiazhuang Tiedao University, grant number KF2021-15, ZZ2021-13.

**Institutional Review Board Statement:** Not applicable.

**Informed Consent Statement:** Not applicable.

**Data Availability Statement:** The data presented in this research can be requested from the corresponding author or the first author.

**Conflicts of Interest:** The authors declare no conflict of interest.

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
