# Peer review of "A Case Study on Structural Serviceability Subjected to Railway-Induced Vibrations at TOD Developed Metro Depot"

_buildings, doi:10.3390/buildings12081070_

Round 1
Reviewer 1 Report
The article is interesting and quite detailed and, in my opinion, can be accepted. It needs, however, a review of the text, to correct spelling and formatting errors.
From a technical point of view, the article can be accepted without changes. However, there are spelling errors in the written text. Proofreading the text, preferably by a native English speaker, is necessary to correct these errors. Also, the formatting (type and size of fonts) needs to be standardised.
Reviewer 2 Report
This manuscript addresses an interesting topic on human comfort problem due to railway induced vibration, which is quite a hot issue raised with the most common public complaints against TOD development. Therefore, the case study provided by this paper to discuss the influence of railway-induced vibration on the neighborhood residents in a TOD developed depot has practical value. Overall, the manuscript is well written and addresses a relevant topic, it would be accepted to be published after minor amendments.
Detailed comments are listed below:
1. The English language should be improved, especially some titles of figures and tables that were not clearly descript their contents. For example, Line 164, Fig.1, maybe "the plan view of the presented depot". And line 165, the mentioned "rail type" would be" train and track parameters".
2. Line 166, in Table 3, regarding to the mentioned "Long sleeper embedded ballastless track" and "Type I separate fastener", but the readers may want to know more detailed technical parameters other than the unfamiliar names.
3. Please provide a detailed description of the data listed in tables, for example, L328, in table 4, what is the unit of the data?
4. This paper provides both the reference standard and measurement data, but the readers may want to know how the measurement data compared to the reference standard, it is possible to clearly describe the comparison together in some added figures and tables?
5. In section 5, this paper mentioned two different evaluation indexes using vibration velocity and acceleration, is it possible to discuss what the difference between the two indexes in this case study?
Reviewer 3 Report
Case Study on Structural Serviceability Subjected to Railway-induced Vibrations at TOD Developed Metro Depot
The article is interesting and well written.
Few minor suggestions are given below;
Abstract need revision with some quantitative results.
Some more latest studies are required in the introduction section to further highlight the importance of this study.
Section 2, further explanations and reasons are required for the selected 11-storey building. Why esetntially this case was considered.
Also, a detail description and references are required for the selected trucks.
Figures 17a and b are totally meaningless because those are not readable at all.
Proper references are required for the equations and also list of abbriations and symbols is required.
Authors must summarized results in more systematic way with reference to the previous studies.
Also, Conclusions are too limited to proof the significant outcome of this study.
